# The Increased Densities, But Different Distributions, of Both C3 and S100A10 Immunopositive Astrocyte-Like Cells in Alzheimer’s Disease Brains Suggest Possible Roles for Both A1 and A2 Astrocytes in the Disease Pathogenesis

**DOI:** 10.3390/brainsci10080503

**Published:** 2020-07-31

**Authors:** Andrew King, Boglarka Szekely, Eda Calapkulu, Hanan Ali, Francesca Rios, Shalmai Jones, Claire Troakes

**Affiliations:** 1Department of Clinical Neuropathology, King’s College Hospital, Denmark Hill, London SE5 9RS, UK; 2London Neurodegenerative Diseases Brain Bank, Institute of Psychiatry, Psychology and Neuroscience, King’s College, London SE5 8AF, UK; Boglarka.szekely@kcl.ac.uk (B.S.); eda.calapkulu@kcl.ac.uk (E.C.); hananyali13@gmail.com (H.A.); francesca.rios@kcl.ac.uk (F.R.); shalmai.jones@kcl.ac.uk (S.J.); claire.troakes@kcl.ac.uk (C.T.); 3Department of Basic and Clinical Neurosciences, Institute of Psychiatry, Psychology and Neuroscience, Kings College, London SE5 9RT, UK

**Keywords:** Alzheimer’s, dementia, immunohistochemistry, A1, A2, astrocytes, glial, C3, S100A10, neurodegeneration

## Abstract

There is increasing evidence of astrocyte dysfunction in the pathogenesis of Alzheimer’s disease (AD). Animal studies supported by human post-mortem work have demonstrated two main astrocyte types: the C3 immunopositive neurotoxic A1 astrocytes and the S100A10 immunopositive neuroprotective A2 astrocytes. A1 astrocytes predominate in AD, but the number of cases has been relatively small. We examined post-mortem brains from a larger cohort of AD cases and controls employing C3 and S100 immunohistochemistry to identify the astrocytic subtypes. There were a number of C3 immunopositive astrocyte-like cells (ASLCs) in the control cases, especially in the lower cerebral cortex and white matter. In AD this cell density appeared to be increased in the upper cerebral cortex but was similar to controls in other regions. The S100A10 showed minimal immunopositivity in the control cases in the cortex and white matter, but there was increased ASLC density in upper/lower cortex and white matter in AD compared to controls. In AD and control cases the numbers of C3 immunopositive ASLCs were greater than those for S100A10 ASLCs in all areas studied. It would appear that the relationship between A1 and A2 astrocytes and their possible role in the pathogenesis of AD is complex and requires more research.

## 1. Introduction

Alzheimer’s disease (AD) affects approximately 47 million people worldwide and is projected to increase to more than 131 million by 2050 [1]. It is the commonest cause of dementia in the Western world, the main risk factors being increasing age, atherosclerosis, diabetes mellitus and family history [2,3,4]. The progressive neurodegenerative disease is characterised neuropathologically by the deposition of Amyloid-Beta (Aβ) plaques and hyperphosphorylated tau in the form of neurofibrillary tangles, neuropil threads and dystrophic neurites. This is associated with synaptic loss and neuronal death [5]. Whilst most research into this disease has concentrated on neuronal dysfunction and the pathophysiological roles of the associated abnormal proteins, more recently, attention has been paid to the contribution of astrocytic activity to the disease progression [6,7,8,9]. Astrocytes are abundant in the central nervous system [10], and their role was considered previously to be primarily supportive of the neuron as acting like a “brain glue” [7,11], but there is now accumulating evidence that, depending on the disease stage, they may be involved in the production of toxic Aβ, synaptic loss and neuronal cell death [8,12,13]. Indeed, studies in animal models of AD have shown that astrocyte activation can actually precede Aβ deposition [14]. Human post-mortem and animal studies have shown persistent numbers of astrocytes in AD cases but a change in morphology and physiology with the disease’s progression [8,15,16]. Discovery of subtypes of macrophages/microglia and their later designation as tissue destructive (M1) or cellular remodelling (M2) posed the question as to whether there were similar subtypes of other glial cells, such as astrocytes [17,18]. The cellular dynamics of AD have been complicated by the issue that it was never clear as to whether the accumulation of microglia and astrocytes in particular areas of the brain were primarily destructive or protective, or depended on the timing in the disease progression pathway. Liddelow et al., whilst using animal models, proposed that there were two main astrocytic states or types [19]. These were A1 astrocytes which were induced by reactive microglia, were neurotoxic and had lost the ability to promote neuronal survival, synaptogenesis and phagocytosis; and A2 astrocytes induced by ischaemic stimuli which upregulated neurotrophic factors and hence were neuroprotective. They determined that certain markers appeared to be A1 astrocyte specific, most notably the complement factor C3, and some were A2 astrocyte specific, such as the calcium binding protein S100A10. They then looked at these markers in human post-mortem brains, most notably with neurodegenerative diseases including AD, amyotrophic lateral sclerosis (ALS), Parkinson’s disease (PD) and Huntington’s disease (HD). They found that A1 astrocytes were abundant and were the predominant astrocytic subtype in all these diseases and in numbers that were much greater than in the controls. However, the number of cases included in each of the disease entities were relatively small and the group primarily employed immunofluorescence. Whilst this allows double or triple immunostaining it is less precise in anatomically localising the exact site of the cells in the brain. Therefore, the purpose of this current investigation was to use the more permanent immunoperoxidase technique in order to examine a larger cohort of AD cases and determine whether we could also detect an increase in these C3 immunopositive A1 astrocytes compared to the S100A10 immunopositive A2 astrocytes and, if so, where these different cell types were located with reference to the associated AD pathology the cerebral cortical layers, and underlying white matter.

## 2. Materials and Methods

The brains from 12 cases of neuropathologically confirmed Alzheimer’s disease were selected from the London Neurodegenerative Diseases Brain Bank. These cases had been clinically confirmed as dementia and neuropathologically had a BNE (Brain Net Europe) modified Braak stage of V or VI [20,21]. Ten of these cases had neuropathological “AD” pathology with no significant cerebrovascular pathology or any co-existing TDP-43 pathology or Lewy Body pathology. Two cases had additional Lewy Body pathology (limbic stage according to McKeith staging) [22]. The brains from 22 control cases were also selected from the London Neurodegenerative Diseases Brain Bank. These were brains from patients with no clinical history of cognitive decline, and a maximum BNE/modified Braak stage of II, with no other significant tau, α-synuclein or TDP-43 pathology or cerebrovascular disease. Where available the following formalin fixed paraffin embedded blocks were used: middle frontal gyrus (Brodmann areas 8/9), middle and superior temporal gyri (Brodmann areas 21/22) and posterior hippocampus with parahippocampal gyrus. Fifteen of the control cases had the full complement of blocks available, four cases had only middle frontal gyrus available, two cases had only the superior and middle temporal gyrus available and one case had only the hippocampal block available. All 12 cases of AD had the full complement of blocks available. In addition, affected cerebral blocks from three cases of old cerebral infarction (of at least 3 weeks prior to death) and two cases of acute infarction (days’ duration—less than one week before death) were selected. These cases were examined to test the hypothesis that A2 astrocytes were more predominant in ischaemia/infarction. Consent for autopsy, neuropathological assessment and research was in place for all cases and the study was carried out under the ethical approval of the brain bank.

### 2.1. Immunohistochemistry

Sections of 7 µm thickness were cut from the paraffin-embedded tissue blocks, deparaffinised in xylene, endogenous peroxidase was blocked by 2.5% H_2_O_2_ in methanol and immunohistochemistry was performed. Details of the antibodies employed are summarised in Appendix A. To enhance antigen retrieval sections were kept in a citrate buffer for 10 min following microwave treatment. After blocking in normal serum (DAKO, Cambridgeshire, UK) a primary antibody was applied overnight at 4 °C. Following washes, the sections were incubated with biotinylated secondary antibody (DAKO), followed by avidin:biotinylated enzyme complex (Vecta-stain Elite ABC kit, Vector Laboratories, Peterborough, UK). Finally, the sections were incubated for 5–10 min with 0.5 mg/mL 3,3′-diamobenzidine chromogen (Sigma-Aldrich Company Ltd., Dorset, UK) in Tris-buffered saline (pH 7.6) containing 0.05% H_2_O_2_. The sections were counterstained with Harris’ haematoxylin and analysed by light microscopy. With both the control cases and the AD cases assessment was made of the overall appearances of the staining within the specified blocks with particular attention to neuropil staining for plaques, any likely glial especially astrocytic-like cell (ASLC) staining and their positioning in the cortex (upper cortex—cortical layers I–III, lower cortex—cortical layers IV–VI) and in the white matter. In the hippocampus the presence of plaques and glial (especially astrocytic—like staining) was assessed in the CA4 region. Furthermore, the presence of any neuronal-type staining was noted, including neurofibrillary tangles. With respect to the plaques and ASCLs, a semi-quantitative score was given. These score parameters were as follows: − negative, +/− only very infrequent, + infrequent, ++ moderate numbers/moderate density, +++ large numbers/high density. In addition, because the frontal lobe blocks were the most uniform in anatomical site and to offer some comparison with the prefrontal blocks examined by Liddelow et al., they were selected for more detailed assessment [19]. Using a two-dimensional assessment, a more objective evaluation than simple semi-quantitation was attempted, although it should be appreciated that because of the nature of the specimens the assessments could not reach the best practice of sterology. Within the frontal cortex and frontal white matter block counts of astrocyte-like cells (ASLCs) were taken. These were counted with an Olympus BX51 microscope using a X40 objective lens (numerical aperture (NA) 0.9) giving a count area of 0.327 mm^2^. The parameters for counting were those cells with cytoplasmic immunopositivity for either C3 or S100A10 with associated distinct nuclei visible and being not obviously neurons, also with a diameter (without processes) of less than 20 µm. All counts were made with the same light intensity, and counts were only recorded as positive if the nucleus of the cell was visible, reducing some bias as to the plane of section. Furthermore, any areas with tissue distortion were excluded. These counts were made on eight cases of “pure” AD and eight control cases and the counts were taken of upper cortex, lower cortex and white matter. Ten field of views were examined per area per case and 10 counts were made in each area for each case. In the frontal upper cortex and lower cortex there were approximately 150 cells surveyed per count and in the white matter approximately 300 cells surveyed per count and the results assessed with the Kruskal-Wallis test with correction and with a significance level of 0.05. Assessments were made comparing the mean count of C3 positive ASLCs in control v AD cases, the mean count of S100A10 positive ASLCs control v AD cases and the mean count of C3 positive ASLCs v the mean count of S100A10 positive ASLCs in the different areas of both control and AD cases.

In the cases with cerebral infarcts these were assessed to determine the degree of respective staining for C3 and S100A10 and the cells that they were likely to be detecting.

### 2.2. Immunofluorescence

Double immunofluorescence was carried out with 7 µm sections cut from formalin fixed paraffin embedded blocks, dewaxed in xylene and dehydrated in 99% industrial methylated spirit. Sections were pre-treated by microwaving in a citrate buffer and blocked using normal goat serum (1:10 for 45 min). Combinations of primary antibodies were then applied, and sections were incubated overnight at 4 °C. The details of these antibodies are shown in Appendix A. Sections were washed and secondary Alexa Fluor antibody (goat anti-mouse 488 and goat anti-rabbit 568, Invitrogen, Paisley, UK) applied for 45 min (in dark). Autofluorescence was quenched by incubating the sections in Sudan black for 10 min followed by numerous washes in phosphate buffered saline before coverslip mounting using hard set media with 4′,6-diamidino-2-phenylindole (DAPI). Sections were visualised using a fluorescent microscope (Zeiss AxioImager Z1, Gottingen, Germany), and images were captured using AxioVision Rel 4.8.2.

## 3. Results

The results for C3 and S100A10 in all regions stained in all the control and AD cases and their associated semi-quantitative scores are summarised in Appendix A.

### 3.1. C3 Immunostaining

#### 3.1.1. Controls

The C3 immunohistochemistry in control cases showed only very occasional staining for what appeared to be ASLCs (Figure 1a) in the upper frontal and temporal cortex. However, in the lower cortex there were variable but often frequent C3 positive ASLCs (Figure 1b). These predominantly appeared to be compact cells and the C3 staining was localised mainly around the nucleus. The cells were also often adjacent to neurons. Within the frontal and temporal white matter there were variable, but often frequent numbers of C3 positive ASLCs, again mainly compact appearing cells with strong perinuclear immunopositivity (Figure 1c). There did not appear to be obvious differences in the density of C3 positive ASLCs in the different regions of cortex and white matter when comparing the frontal and temporal lobe. In the hippocampus in the CA4 region there were occasional C3 immunopositive ASLCs present with some cases showing more frequent positive cells (Figure 1d).

#### 3.1.2. AD Cases

In many of the AD cases there was C3 immunostaining in the pattern of plaques (Figure 2a). This appeared to be neuropil staining and was present in varying densities in the cortex (usually upper cortex) of both the frontal and temporal lobes. In addition, there were what appeared to be increased numbers of C3 immunopositive ASLCs in the upper frontal and temporal cortex compared to controls (Figure 2a–c). Within the lower frontal and temporal cortex there were variable and often frequent numbers of C3 immunopositive ASLCs (Figure 2d). The white matter revealed usually frequent C3 immunopositive ASLCs (Figure 2e). Sometimes the distribution was focal and variable. In the hippocampal CA4 region there were usually C3 immunopositive distinct plaques demonstrated (Figure 2f), and variable numbers of separate C3 immunopositive ASLCs (Figure 2g). In occasional AD cases the C3 appeared to be labelling neurofibrillary tangles in addition to the glial staining (Figure 2h).

The statistical analysis performed on results from the frontal lobe of control and AD cases showed a significantly greater number of C3 immunopositive ASLCs in the upper cortex of AD cases compared to controls (Figure 3). There was, however, no significant difference detected between the number of C3 immunopositive ASLCs in the frontal lower cortex (Figure 3) and white matter of the AD cases and controls (Figure 3). (See Appendix A for statistical counts). There appeared to be no obvious difference in the density and distribution of the C3 immunopositivity between the “pure AD” cases and those two AD cases with additional limbic Lewy Body pathology.

### 3.2. S100A10 Immunostaining

#### 3.2.1. Controls

The S100A10 immunostaining in control cases showed very little, if any, staining in the upper or lower cortical layers of the frontal and temporal lobes apart from blood vessels (Figure 4a,b). Similarly, in the frontal and temporal white matter of control cases there were usually very few, if any, S100A10 immunopositive ASLCs seen. Those that were seen were often adjacent to blood vessels (Figure 4c). Within the hippocampus again there were usually very few, if any S100A10 immunopositive ASLCs cells seen in the CA4 region (Figure 4d).

#### 3.2.2. AD Cases

In AD cases in the frontal and temporal cortex S100A10 immunopositive plaques were identified (Figure 5a). These were, however, usually less apparent that those seen with the C3 immunostaining. In addition, there were, in both the upper and lower frontal and temporal cortex, moderate numbers of S100A10 immunopositive ASLCs present (Figure 5a–d). These were often star-shaped in appearance. Some, but not all, were seen immediately adjacent to plaque-like structures. There were also usually moderate numbers of S100A10 immunopositive ASLCs in the white matter of the frontal (Figure 5e) and temporal lobes. Within the hippocampal CA4 region there were often S100A10 immunopositive plaques (Figure 5f) and moderate numbers of S100A10 immunopositive ASLCs (Figure 5 g). In occasional AD cases the S100A10 appeared to be labelling neurofibrillary tangles in addition to the ASLC staining (Figure 5h). The statistical analysis performed on the frontal lobe of control and AD cases showed a significantly increased number of S100A10 immunopositive ASLCs in the upper and lower frontal cortex and white matter of AD cases compared to controls (Figure 6).

In the AD cases there were significantly increased numbers of C3 immunopositive ASLCs in the frontal upper cortex, lower frontal cortex and white matter when compared to S100A10 cases, but the upper frontal cortex showed the least significance (Figure 7). However, in the control cases there were also significantly increased numbers of C3 immunopositive ASLCs in the frontal upper cortex, lower frontal cortex and white matter when compared to S100A10 cases (Figure 8). There appeared to be no obvious difference in the density and distribution of the S100A10 immunopositivity between the “pure AD” cases the two AD cases with additional limbic Lewy Body pathology.

### 3.3. Immunofluorescence

The double immunostaining in AD cases showed that C3 and S100A10 appeared to stain different glial cell types (Figure 9a). Furthermore, there appeared to be no co-expression of C3 and glial fibrillary acidic protein (GFAP) in the glial cells of the samples studied (Figure 9b), whereas there was at least some S100A10 immunopositive co-expression of GFAP (Figure 9c,d).

### 3.4. Infarction Cases

Within the acute infarct cases there appeared to be expression of C3 and S100A10 (Figure 10a,b), although it was not always apparent which cells were expressing the markers. It appeared to be mainly ASLCs, but in some adjacent regions neurons appeared to be positive (Figure 10b). Within the old infarcts there were only occasional weakly stained C3 immunopositive ASLCs identified (Figure 10c), but stronger immunostaining was seen for S100A10 in ASLCs (Figure 10d). Away from the infarcted area neurons were seen occasionally expressing C3 and in other areas S100A10 (Figure 10e,f). The macrophages within the infarcts appeared to be negative both for C3 and S100A10.

## 4. Discussion

AD is the commonest cause of neurodegeneration in the west, and whilst historically researchers have concentrated on the direct toxicity of Aβ and hyperphosphorylated tau on the neurons, recently much more attention has been paid to the role microglial cells and astrocytes play in neuronal dysfunction and loss [6,7,8,13]. Considering that astrocytes are an abundant cell type in the CNS and their role has been thought to be very important in metabolic support and the protection of neurons [11], it is now being appreciated that astrocytic dysfunction may have a primary role in the pathogenesis of neurodegenerative diseases [9]. Indeed, it is now known that astrocytes can convert pyruvate to lactate which is shuttled to neurons (which themselves rely on oxidative metabolism) where it can be used as an energy substrate—the so-called astrocyte-neuron lactate shuttle—and, therefore, any dysfunction in astrocytes is very likely to also adversely affect neurons [23,24]. Whereas the original amyloid cascade model of AD progression by Hardy et al., suggested abnormal Aβ production mainly being by neurons there has been accumulating evidence incriminating astrocytes as a significant co-producer of the protein [12,25]. In the last few years there has been some evidence to implicate different forms of microglia in different pathogenic conditions, namely M1 pro-inflammatory microglial cells and M2 protective or anti-inflammatory microglial cells, although there have been suggestions that this concept is overly simplified and it is not universally accepted [17,18,26]. Following on from the assumption that there is, however, some type of microglial differentiation, there have been additional experiments using animal models that have provided evidence of a similar subdivision of astrocytes. Some studies have concluded that astrocytes could be divided into neurotoxic so-called A1 astrocytes (induced by reactive microglia) and neuroprotective so-called A2 astrocytes [19,27,28,29]. Using animal models and confirming with human brain material Liddelow et al., showed that the so called A1 astrocytes lose the ability to promote neuronal survival, synaptogenesis and phagocytosis, and appear to induce the death of neurons, whereas so called A2 astrocytes upregulate many neurotrophic factors and were more likely to be neuroprotective [19]. They found that certain markers were A1 astrocyte specific and some were A2 astrocyte specific, and determined immunocytochemically that a complement component called C3 (using an antibody against the cleavage product C3d) specifically labelled A1 astrocytes, whereas the calcium binding protein S100A10 (also called p11) was A2 astrocyte specific. They showed in the human central nervous system that A1 astrocytes were the predominant astrocytic subtype in neurodegenerative diseases such as AD, ALS, HD and PD as well as multiple sclerosis (MS) [19]. Within AD cases they also demonstrated with in situ hybridisation that in the prefrontal cortex and hippocampus C3 positive (A1 astrocytes) were the predominant astrocytes and seen in much greater numbers than in controls. It was also postulated from animal experiments that the A2 astrocytes were particularly important in cerebral ischaemia and infarction [19]. Other studies have shown A1 astrocytes to be important in PD, prion disease and head trauma [27,29,30]. Our preliminary findings on human post-mortem material offer some support to the idea that the S100A10 positive astrocytes may have a more predominant role in old cerebral infarcts, although it is less obvious in early infarcts where both the C3 and S100A10 appear to be expressed. Our study using immunohistochemistry in AD and control brains, however, implied a somewhat more complex pathological pattern than A1 (C3 immunopositive) astrocytes having the main pathogenic role and A2 (S100A10 immunopositive) astrocytes being much less important. Firstly, and not surprisingly, our study revealed that C3, which is part of the complement pathway, is not an entirely specific marker for astrocytes, as it was occasionally seen here in neurons (including tangles) and in the neuropil associated with plaques. Similarly, S100A10 was seen here occasionally in neurons including tangles. Nevertheless, the majority of the cells staining for C3 and S100A10, respectively, did appear to have characteristic features of astrocytes. Liddelow et al., appeared to correct for this possible lack of cell-type specificity with immunofluorescent co-staining for C3 and GFAP (or S100β) [19]. Our double immunofluorescence staining revealed that the combined C3 and S100A10 immunostaining labelled two different astrocyte-like cell populations. The technique also showed that the astrocytic-like C3 immunopositive cells did not co-express GFAP, but the S100A10 did show a proportion of ASLCs that co-expressed GFAP. Although initially this may appear surprising is not necessarily a contradiction to the concept of the cells being astrocytes. It has been known for many years that the Alzheimer type II astrocytes seen in hepatic encephalopathy are GFAP negative, and previous studies have shown that astrocytes are not all necessarily GFAP positive and, indeed, it may be that they only express this protein in certain physiological or pathological states [15,31,32,33,34,35]. Furthermore, when double immunofluorescence for C3 and a microglial marker CD68 had been attempted previously there was also no evidence of co-expression (Calapkulu—personal communication) reducing the possibility here that the C3 is actually staining for microglia. There has been interesting work on the role of the brain water channel Aquaporin 4 (AQP4) in astrocytes. It is highly localised on the endfeet of astrocytes under normal conditions and has a role in neuroprotection in AD [36,37]. In reactive astrocytes it is seen in the cytoplasm as well as endfeet, and there is evidence that it may be involved in astrocyte—microglial communication [36]. It, therefore, may be a good marker for GFAP negative reactive astrocytes in future studies. Given that a high proportion of the C3 positive cells we detected were likely to be astrocytic from their overall appearance, it is particularly intriguing that in control cases there appeared to be a relatively abundant population of C3 positive ASLCs present presumably in a “quiescent” form, both within the lower cortex and the cerebral white matter. Interestingly, although not specifically commented upon in their paper (but seen in their Appendix A) Hartmann et al., appeared to show similar features in the cerebrum of control cases [30]. If the C3 immunopositive ASLCs do, as Liddelow et al., suggest, represent neurotoxic A1 astrocytes, what could possibly be their physiological role in a supposed “quiescent” state? Could there be a further subdivision needed of A1 astrocytes into resting or active, or could they be potentially anti-pathogen related or employed in a regulatory role in the remodelling of axons? Interestingly, very recent work by Bayraktar and colleagues provided evidence of diversity of astrocytic features across cerebral cortical layers and these differences being due to cues from local neurons [38]. It may, therefore, be that the level of “distress” experienced by neurons could dictate the characteristics and functions of neighbouring astrocytes. In our study the numbers of C3 immunopositive ASLCs in the frontal lower cortex and white matter were actually not significantly different between AD and control cases. It was only in the upper cortex (cortical layers I–III) where a significant increase of the C3 positive ASLCs could be seen in AD cases compared to controls. It is probably more likely in AD that the C3 positive A1 astrocytes have differentiated from a pre-existing different glial/astroglial subtype or glial stem cell [34,39], rather than migrated from deeper in the cortex or white matter. Interestingly, there is evidence that the density of A1 astrocytes increases in ageing [28]. Since ageing is one of the most important risk factors for developing AD it may be that the A1 astrocytes increase in density in age, and, perhaps, there is also a greater propensity of precursor cells to convert to A1 astrocytes in older patients given the correct stimuli such as Aβ or hyperphosphorylated tau deposition. However, in our series although small in number, five of the control cases with frontal and/or temporal blocks were aged 25–51, and they did not obviously demonstrate fewer C3 immunopositive ASLCs in the deep cortex or white matter when compared to the older aged control cases. In our study the AD cases showed statistically significant greater numbers of S100A10 immunopositive ASLCs in the frontal upper cortex, lower cortex and white matter when compared with controls. If, as would seem likely, the majority of these S100A10 positive ASLCs corresponded to the A2 astrocytes suggested by Liddelow et al., then the findings imply that these cells were involved in the pathogenesis of AD notwithstanding their supposed neuroprotective role [19]. Within the literature it is noteworthy that the A2 subtype of the astrocyte has seemingly elicited less interest when compared to the A1 [40,41]. The results do not necessarily contradict the findings of Liddelow et al., since when comparing actual numbers of C3 immunopositive ASLCs and S100A10 immunopositive ASLCs per area in the frontal lobe of AD cases our results still showed a statistically higher number of C3 immunopositive ASLCs in all areas. The flipside to this finding, however, is that there were also more C3 immunopositive ASLCs than S100A10 immunopositive ASCLs in all areas of the control cases. From the fluorescence results it appears that C3 and S100A10 immunostaining are confirming two different cell populations. Therefore, under “normal” conditions there would appear to be greater numbers of the C3 immunopositive ASLCs than S100A10 immunopositive ASLCs in the upper and lower frontal cortex and the frontal white matter. During the pathogenesis of AD, however, there is an increase in the number of C3 positive ASLCs evident in the upper cortex. There is also an increased number of S100A10 positive ASLCs in the frontal upper frontal cortex, lower cortex and white matter, but they are still outnumbered in all areas by the C3 positive cells. This may possibly indicate that in AD following certain stimuli there is a “neurodestructive” process occurring via A1 astrocyte upregulation and activation primarily in the upper cortex and facilitated by the numbers of pre-existing “quiescent” A1 astrocytes (which in itself is possibly exacerbated with increasing age) followed by an upregulation and activation of A2 astrocytes in all areas in an attempt perhaps at belated neurotropic support and/or axonal repair. The concept of both a destructive and reparative role of astrocytes depending on the time course of a disease has been discussed by Zhou et al., with reference to traumatic brain injury [42]. Although the idea of a straightforward A1-toxic astrocyte role, and an A2 supportive role, is supported by studies in head trauma and PD [27,41], it has been somewhat complicated by the work of Hartmann et al. [30]. They showed that C3 immunopositive, A1 astrocytes were abundant in the brains of mouse models of prion disease but that surprisingly their abolishment led to an accelerated disease process, possibly by interfering with microglial functioning that was otherwise slowing the disease progression [30]. Therefore, the inter-relationship between different subtypes of microglial cells and different subtypes of astrocytes may be significantly more complex than a simple balance between neurotoxic and neurosupportive cells. The most obvious drawback to our study is the fact that we have not comprehensively proven that the C3 and S100 immunopositivity exhibited in glial cells was highlighting astrocytes, although there was some supportive evidence in the double immunofluorescence at least with respect to S100A10 immunopositive cells. Similarly, we were relying on one marker each for the so called A1 and A2 astrocytes, and even though these would appear to be the most robust markers relied on by Liddelow et al., in their study on mice and human brains [19], it would nevertheless be useful to employ other markers such as AQP4 to confirm the consistency of immunolabelling of these putative astrocytic subtypes. If a further panel of such antibodies could be found that were reliable on formalin-fixed, paraffin-embedded material then not only could cases of AD be further evaluated but also so could other neurodegenerative diseases. It could thus be better established as to the relative make up and likely role of the different glial cell types in the respective neurodegenerative diseases. This would be a necessary precondition before considering therapeutic intervention by way of specific astrocyte promotion or attenuation in any subtype of neurodegenerative disease.

## 5. Conclusions

In earlier animal and some human studies the complement factor C3 has been shown to be a marker of A1 “neurotoxic astrocytes”, whereas S100A10 was shown to label A2 “neuroprotective” astrocytes. Employing immunohistochemistry we have demonstrated that in the frontal lobe of AD brains there is an increase of both C3 and S100A10 immunopositive astrocyte-like cells (ASLCs) compared with controls, but in the case of C3 this increase is only evident in the upper frontal cortex whereas with S100A10 it is seen in frontal upper and lower cortex and white matter. Nevertheless, in all areas the C3 immmunopositive ASLCs remain more numerous than the S100A10 immmunopositive ASLCs. This therefore suggests that both A1 and A2 astrocytes may play roles in the pathogenesis of AD.

## Figures and Tables

**Figure 1 brainsci-10-00503-f001:**
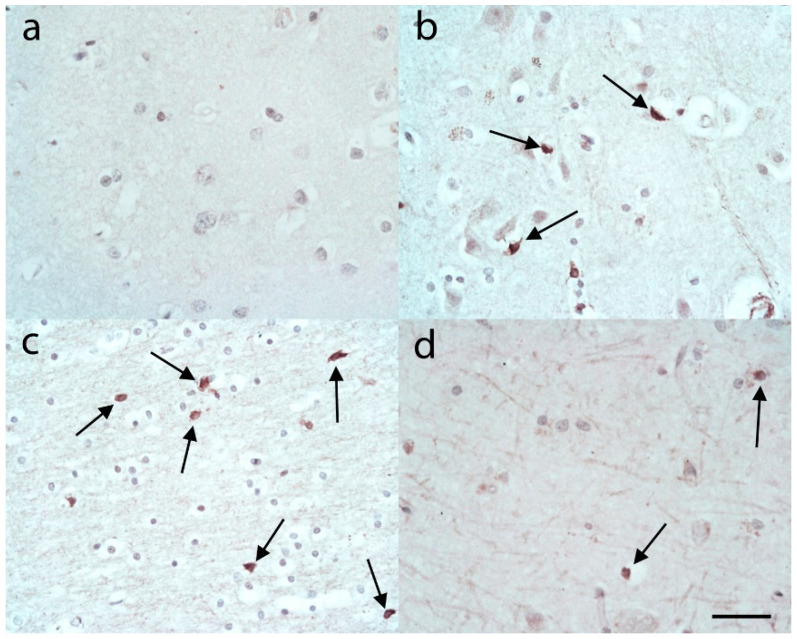
C3 immunohistochemistry in control cases revealed very few or no astrocyte-like cells (ASLCs) in the upper frontal cortex (**a**). There were, however, often moderate or sometimes abundant ASLCs present in the lower frontal cortex (**b**, arrows) and frontal white matter (**c**, arrows). Occasional ASLCs were seen in the CA4 region of the hippocampus (**d**, arrows). Anti-C3d. Original magnifications (**a**–**d**) ×40 (numerical aperture (NA) 0.9). Scale Bar (**a**) −60 µm, (**b**–**d**) −50 µm.

**Figure 2 brainsci-10-00503-f002:**
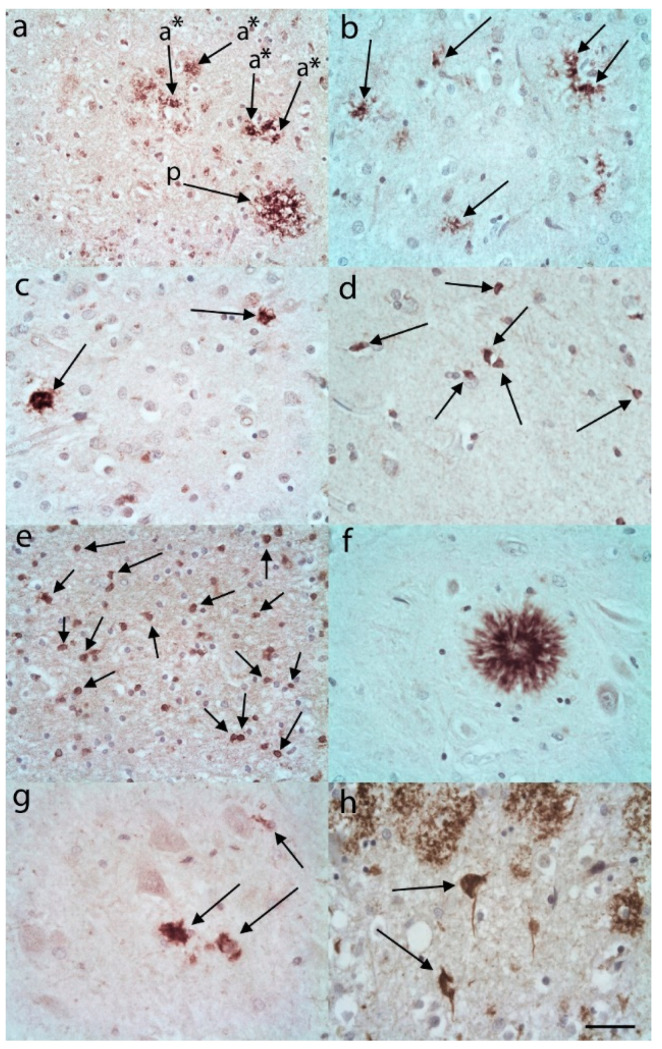
C3 immunohistochemistry in Alzheimer’s disease cases. (**a**) In the upper frontal cortex plaques (p, arrows) and numerous ASLCs (a *, arrows) were labelled. (**b**) There were also numerous ASLCS not adjacent to plaques in the upper frontal cortex (arrows). (**c**) These C3 immunopositive ASLCs were also evident in the upper temporal cortex (arrows). (**d**) The lower frontal cortex revealed moderate to large numbers of C3 immunopositive ASLCs (arrows) and they were numerous in the frontal white matter (**e**, arrows). (**f**) Well formed C3 immunopositive plaques were evident in the CA4 region of the hippocampus. There were also a number of C3 immunopositive ASLCs in the CA4 region (**g**, arrows). (**h**) C3 immunohistochemistry labelled occasional neurofibrillary tangles (arrows). Anti-C3d. Original magnifications (**a**,**d**,**e**,**f**) ×40 (NA 0.9), (**b**,**c**,**g**,**h**) ×60 (NA 0.9). Scale Bar (**a**) −100 µm, (**b**) −60 µm, (**c**,**d**,**f**,**h**) −50 µm, (**e**) −40 µm, (**g**) −30 µm.

**Figure 3 brainsci-10-00503-f003:**
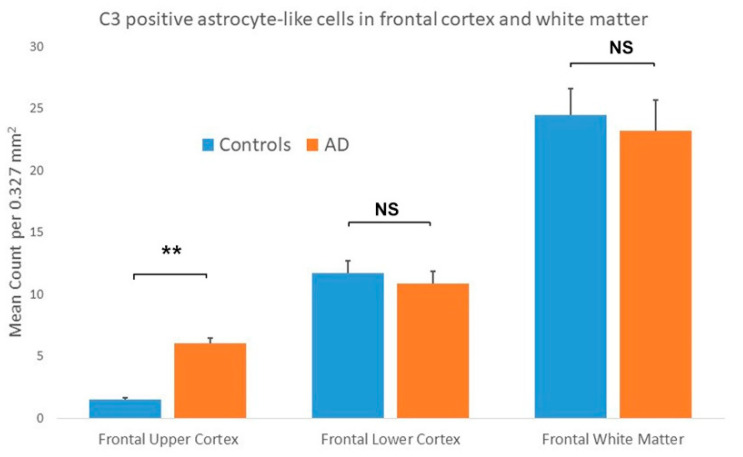
Results showing the mean counts of C3 immunopositive astrocyte-like cells (ASLCs) in the upper frontal cortex, lower frontal cortex and frontal white matter of controls and Alzheimer’s disease (AD) cases. There were significantly increased numbers of ASCLs seen in the upper frontal cortex of AD cases compared to controls but no significant differences in the lower frontal cortex and frontal white matter (** *p* < 0.001, NS-not significant). Results are mean counts + SEM.

**Figure 4 brainsci-10-00503-f004:**
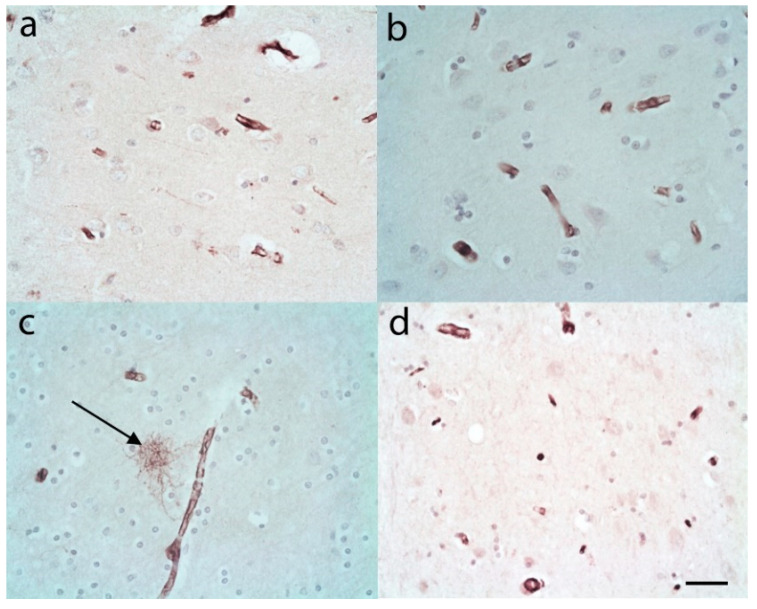
S100A10 immunohistochemistry in control cases revealed positivity in vessel walls but extremely low numbers or no astrocyte-like cells (ASLCs) in the upper frontal cortex (**a**), and the lower frontal cortex (**b**). Only very occasional S100A10 (usually perivascular) immunopositive ASLCs (arrow) were evident in the frontal white matter (**c**). (**d**) The CA4 region of the hippocampus often also revealed no S100A10 immunopositive ASLCs. Anti-S100A10. Original magnifications (**a**–**d**) ×40 (NA 0.9). Scale Bar (**a**,**b**) −60 µm, (**c**) −40 µm, (**d**) −50 µm.

**Figure 5 brainsci-10-00503-f005:**
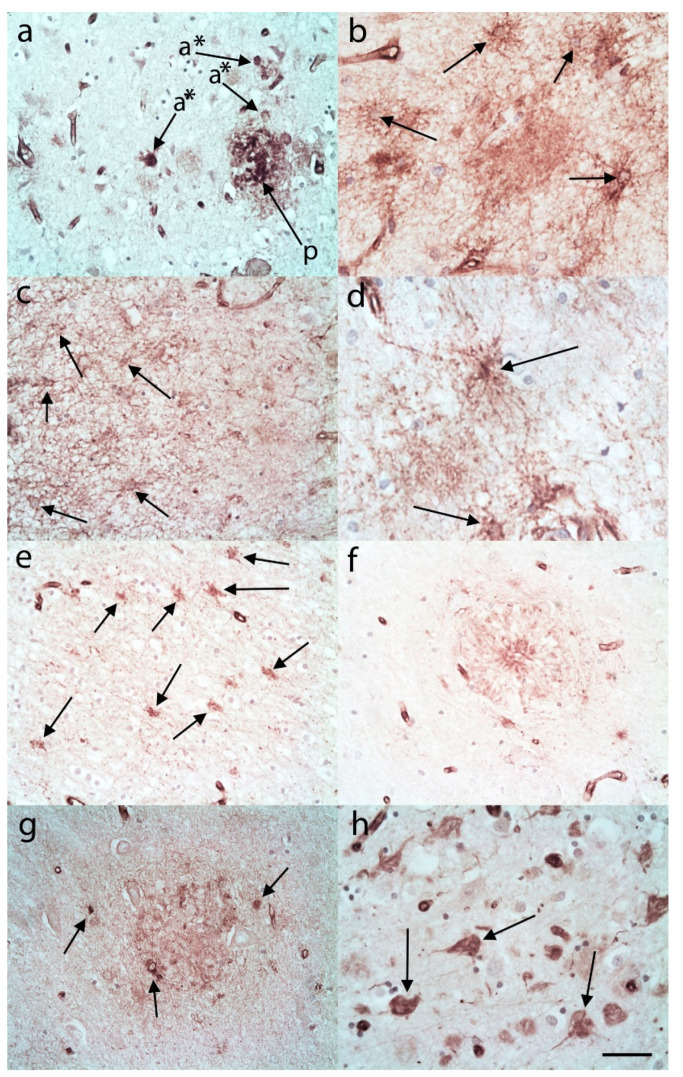
S100A10 immunohistochemistry in Alzheimer’s disease cases labelled in (**a**) the upper frontal cortex plaques (**p**, arrows) and numerous ASLCs (**a ***, arrows). (**b**) There were also numerous S100A10 immunopositive ASLCS not adjacent to plaques in the upper frontal cortex (arrows). (**c**) These S100A10 immunopositive ASLCs were also evident in the upper temporal cortex (arrows). (**d**) The lower frontal cortex revealed moderate to large numbers of S100A10 immunopositive ASLCs (arrows) and they were numerous in the frontal white matter (**e**, arrows). Occasional S100A10 immunopositive plaques were evident in the CA4 region of the hippocampus (**f**). There were variable numbers of S100A10 immunopositive ASLCs also seen in this CA4 region of the hippocampus (**g**, arrows). (**h**) S100A10 immunohistochemistry labelled occasional neurofibrillary tangles (arrows). Anti-S100A10. Original magnifications (**a**–**e**) and (**g**) ×40 (NA 0.9), (**f**,**h**) ×60 (NA 0.9). Scale Bar (**a**) −75 µm, (**b**,**d**,**e**,**g**) −40 µm, (**c**,**f**,**h**) −50 µm.

**Figure 6 brainsci-10-00503-f006:**
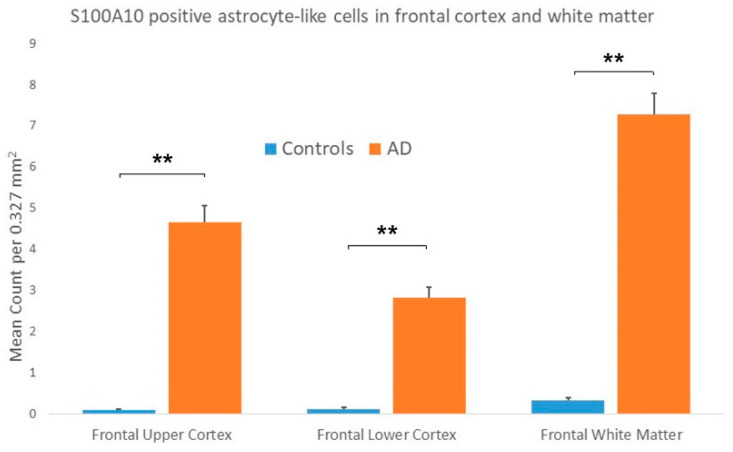
Results showing the mean counts of S100A10 immunopositive astrocyte-like cells (ASLCs) in the upper frontal cortex, lower frontal cortex and frontal white matter of controls and Alzheimer’s disease (AD) cases. There were significantly increased numbers of ASCLs seen in the upper frontal cortex, lower frontal cortex and frontal white matter of AD cases compared to controls. (** *p* < 0.001) Results are mean counts + SEM.

**Figure 7 brainsci-10-00503-f007:**
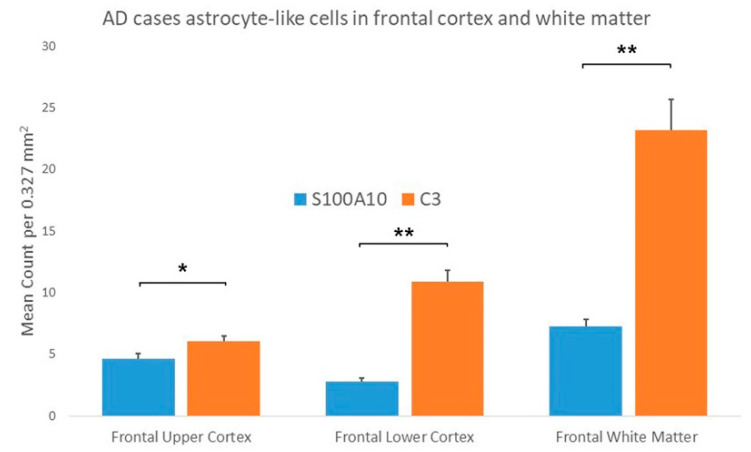
Results comparing the mean counts of C3 and S100A10 immunopositive astrocyte-like cells (ASLCs) in the upper frontal cortex, lower frontal cortex and frontal white matter of AD cases. There were significantly increased numbers of C3 immunopositive ASCLs compared to S100A10 immunopositive ASLCs in all regions but there was least difference in the upper frontal cortex. (* *p* < 0.01, ** *p* < 0.001) Results are mean counts + SEM.

**Figure 8 brainsci-10-00503-f008:**
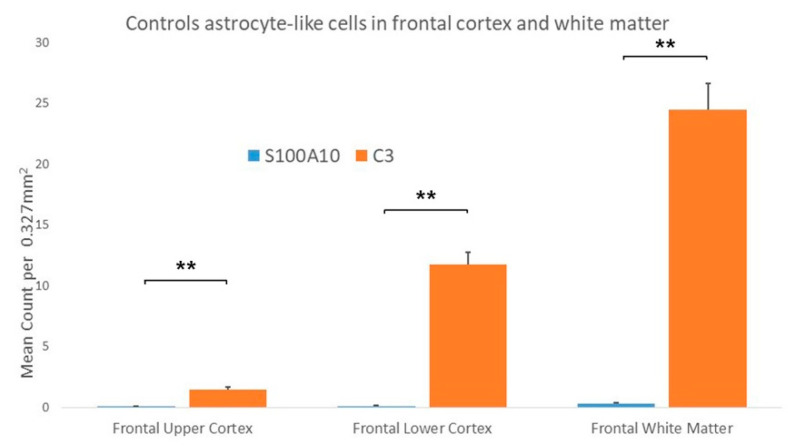
Results comparing the mean counts of C3 and S100A10 immunopositive astrocyte-like cells (ASLCs) in the upper frontal cortex, lower frontal cortex and frontal white matter of control cases. There were significantly increased numbers of C3 immunopositive ASCLs compared to S100A10 immunopositive ASLCs in all regions. (** *p* < 0.001) Results are mean counts + SEM.

**Figure 9 brainsci-10-00503-f009:**
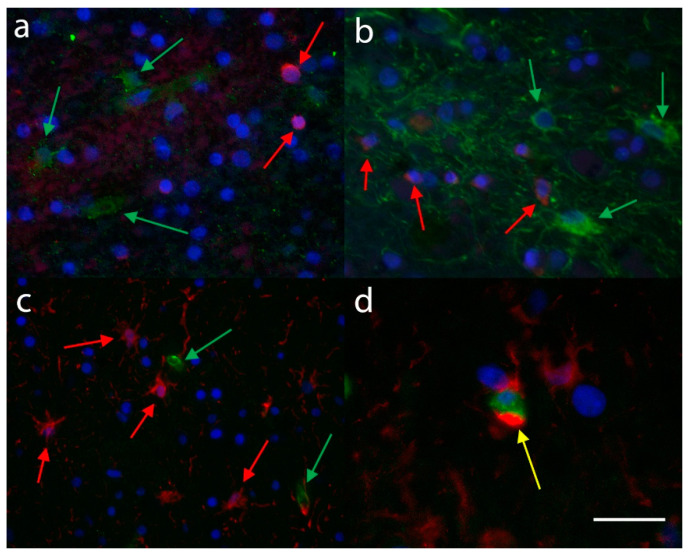
The double immunofluorescence in AD cases revealed (**a**) the C3 immunopositive ASLCs (red, arrows) to be a different population from the S100A10 immunopositive ASLCs (green, arrows) with no cells showing co-localisation. In (**b**) there appeared to be no co-localisation between the C3 immunopositive ASLCs (red, arrows) and the GFAP immunopositive ASLCs (green, arrows). Whereas many of the S100A10 immunopositive ASCLs (green, arrows) did not co-localise with GFAP immunopositive ASLCs (red, arrows) (**c**), there were a number of ASLCs that did co-localise (**d**, arrow). Scale Bar (**a**,**b**) −50 µm, (**c**) −75 µm, (**d**) −25 µm.

**Figure 10 brainsci-10-00503-f010:**
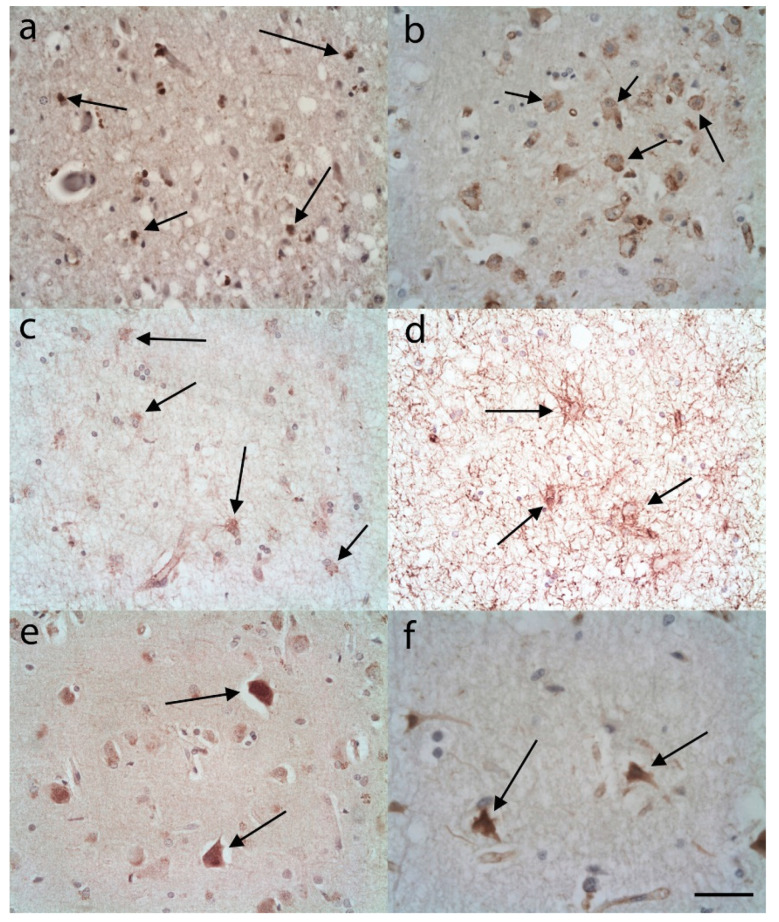
C3 immunohistochemistry (**a**,**c**,**e**); and S100A10 immunohistochemistry (**b**,**d**,**f**); In acute/recent cerebral infarcts (**a**) and (**b**) there were C3 immunopositive cells many of which appeared to be ASLCs (**a**, arrows). There were also a number of S100A10 immunopositive cells some of which appeared to be ASLCs, but some, as shown here, appeared to be neurons (**b**, arrows). In old cerebral infarcts (**c**) and (**d**) there were occasional weakly staining C3 immunopositive ASCLs (**c**, arrows) some of which were star shaped. There were more intensely stained S100A10 immunopositive ASLCs seen in the old infarcts (**d**, arrows). With C3 immunohistochemistry (**e**) and S100A10 immunohistochemistry (**f**) there was occasional neuronal immunopositivity distant from the main focus of infarction (arrows). Anti-C3 (**a**,**c**,**e**); Anti-S100A10 (**b**,**d**,**f**). Original magnifications (**a**–**d**) ×40 (NA 0.9), (**e**,**f**) ×60 (NA 0.9). Scale Bar (**a**–**d**) −60 µm, (**e**,**f**) –50 µm.

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
