# Peer review of "The Increased Densities, But Different Distributions, of Both C3 and S100A10 Immunopositive Astrocyte-Like Cells in Alzheimer’s Disease Brains Suggest Possible Roles for Both A1 and A2 Astrocytes in the Disease Pathogenesis"

_brainsci, 2020, doi:10.3390/brainsci10080503_

Round 1
Reviewer 1 Report
Dear authors,
In their manuscript "Alzheimer's disease brains show increased densities of both C3 and S100A10 immunopositive astrocyte-like cells but with different distributions, suggesting possible role for both A1 and A2 astrocytes in the disease pathogenesis" show the new role of A1 and A2 astrocytes in the AD pathogenesis. Although, the idea behind this research is really interesting and the authors have a lot and really good results. The manuscript has to be improved, basically, in two ways: improving the introduction, material, results and discussion by reorganizing the information presented in the discussion that is for introduction, mainly the first paragraph. Reducing the material methods at least the part of the samples adding a table of the samples to clarify and summarize. Also, the first immunochemistry section is too long.
In general, you have to check the English language and acronyms.
Regarding the title is too long. Consider reducing because is more like a sentence than a title.
Regarding the results:
-Consider modifying the writing of the results in order to explain only the significant results, there is too much useless information. Furthermore, there is repetitive information in the Figure legend of each IHC. There are figure legends without scale bar.
-Consider improving the image quality of the histograms, using another program. Furthermore, introduce the significance like a symbol as well.
Regarding the discussion: Is too long and there's information that you have to add in the introduction. Compare with other studies.
There are more things, please revise in general.
Reviewer 2 Report
Authors take the precaution to talk about astrocyte like cells. Even if these cells are Gfap negative, which is weird because reactive astrocytes generally overexpress Gfap in particular astrocytes proximal to AB deposits, they must find a way to characterize them using coimmunostaining approaches. In addition, C3 is a secreted protein, while the labelling here is restricted to the soma. Another thing is than the labelling of these A1 and A2 markers have already be documented in other studies.
Reviewer 3 Report
Dear editor
The manuscript by King et al investigates the potential role for A1 and A2 astrocytes in the pathogenesis of Alzheimer’s disease (AD).
The design of the study and the technical quality of the work look convincing and results can be of general interest. Authors have successfully managed to discuss the findings of their study through an unbiased comparison with a good range of up-to-date literature. It was good that the authors have already identified some major limitations for this study such as that they have not comprehensively proven that the C3 and S100 immunopositivity exhibited in glial cells was highlighting astrocytes and that they were only relying on one marker each for the so called A1 and A2 astrocytes; which all should be addressed in future studies before driving a solid conclusion on the differential role of these cells in AD pathophysiology.
There is a number of major and minor points that would need to be addressed in order to improve the quality of this paper before it can be accepted for publication:
General:
-Define abbreviations whenever they appear first in the manuscript and use them throughout. For example; line 299 and line 308, where it should be AD since it has been defined earlier.
Major:
-Authors mentioned statistical analysis and drove conclusion based on these findings but there was no mention about how this has been done in the manuscript. Authors need to clearly indicate the statistical approach which should be either student t-test or one-way ANOVA followed by Bonferroni correction or Kruskal–Wallis followed by Conover-Inman post hoc correction.
-Imaging was an essential aspect of this manuscript. Authors need to provide more details such as how many FOVs have been taken, number of cells per every figure, what are their measures to minimize biases, and how they have excluded any possible interference from background signals in order to enhance the reproducibility of the presented data.
- Magnification number should be included for all the figures. But it won’t be enough as it has nothing to do with resolution especially for the purpose of quantitative analyses like in this study. So, authors need to include NA of the utilized lens.
-Figures were poorly presented and it’s hard to read the axes. Axes need to be on the same scale within one figure so readers can have a better understanding when they make comparisons. There should be a better way to include the p values on the figures and it needs to be consistent across all the figures.
-The first part of discussion was very poorly written. It’s simply a copy and paste from their introduction. Consider deleting lines 311-328 as well.
Minor:
-Introduction line 34: it will be good to add some recent epidemiological studies and a reference at the end of this sentence.
-Discussion lines 305-306: authors touched on an important point regarding the important role of astrocytes in metabolic support. They should expand this part and discuss more their role in the brain energy metabolism. References to be included:
-https://pubmed.ncbi.nlm.nih.gov/22152301/
-https://pubmed.ncbi.nlm.nih.gov/31318452/
-Discussion lines 353-354: the results regarding not all the investigated astrocytes were GFAP positive can look a bit unexpected but astrocytes are not all necessarily GFAP positive indeed as they have mentioned. Authors need to discuss considering other biomarkers such as AQP4. This has been nicely shown by the work of Kitchen et al Cell 2020. The parallel expression of GFAP and AQP4 might be a good marker, not only for differentiated astrocytes or reactive astrocytes, but also for precursor cells. References to be included:
-https://www.ncbi.nlm.nih.gov/pmc/articles/PMC7242911/
-https://www.ncbi.nlm.nih.gov/pmc/articles/PMC5000703/
Best.
Round 2
Reviewer 2 Report
I have no other comment.
Reviewer 3 Report
Dear editor
I would like to thank the authors for their efforts to revise the manuscript in the light of the raised concerns and suggestions. The vast majority of my comments have been addressed by the authors accordingly.
The newly added sections, provided references and enhanced figures have helped towards the improvement of the current version compared to their earlier submission.
I would like to recommend this manuscript for publication at Brain Sciences.
Best